# Genetic determinants of pOXA-48 plasmid maintenance and propagation in *Escherichia coli*

Yannick Baffert[1], Nathan Fraikin[1], Yasmine Makhloufi[1], Julie Baltenneck[2], Marie-Eve Val [3], Annick Dedieu-Berne[1], Jonathan Degosserie [4], Bogdan I. Iorga [5], Pierre Bogaerts[6], Erwan Gueguen[2], Christian Lesterlin [1] & Sarah Bigot [1]

Conjugative plasmids are the main drivers of antibiotic resistance dissemination contributing to the emergence and extensive spread of multidrug resistance clinical bacterial pathogens. pOXA-48 plasmids, belonging to the IncL group, emerge as the primary vehicle for carbapenem resistance in *Enterobacteriaceae*. Despite the problematic prevalence of pOXA-48, most research focus on epidemiology and genomics, leaving gaps in our understanding of the mechanisms behind its propagation. In this study, we use a transposon sequencing approach to identify genetic elements critical for plasmid stability, replication, and conjugative transfer. Our results identify a novel type I toxin-antitoxin system, uncharacterized essential maintenance factors, and components of the type IV secretion system and regulatory elements crucial for conjugation. This study advances our understanding of pOXA-48 biology, providing key insights into the genetic factors underlying its successful maintenance and spread in bacterial populations.

Conjugative plasmids are extrachromosomal DNA elements responsible for the spread of antimicrobial resistance (AMR) through the transfer of resistance genes across bacterial populations. Multidrug-resistant commensal, environmental, and pathogenic strains have revealed a wide range of conjugative plasmids carrying genes that confer resistance to nearly all antibiotics currently used in clinical treatments. Carbapenem resistance in *Enterobacteriaceae* has emerged as a critical public health concern, largely attributed to the IncL group pOXA-48 plasmids. These plasmids are the main carriers of the $bla_{OXA-48}$ gene, which encodes the OXA-48 carbapenemase within the Tn*1999* transposon[1].

Despite the rapid global dissemination of pOXA-48 among enteric pathogens, previous studies have primarily focused on epidemiological tracking and genomic characterization, leaving the mechanistic aspects of its transfer and persistence poorly defined. Recent works have shed light on the plasmid dynamics of pOXA-48 in different bacterial hosts, revealing variability in fitness costs and conjugation across strains[2-4]. This variability suggests that plasmid persistence and transfer are influenced by host diversity, evolutionary pressures, and complex genetic interactions[5,6]. Consistent with these findings, a recent study discovered the peculiarity of the *Escherichia coli* ST38 strain to integrate the $bla_{OXA-48}$ gene into its chromosome, driven by the fitness cost associated with the pOXA-48 plasmid and its destabilization by a recent characterized antiplasmid system, ApsAB[7]. Altogether, these results suggest that understanding genetic interactions between plasmids and hosts is key to predicting successful plasmid maintenance and spread.

[1]Microbiologie Moléculaire et Biochimie Structurale (MMSB), Université Lyon 1, CNRS, Inserm, UMR5086, Lyon, France. [2]Microbiologie Adaptation et Pathogénie (MAP), Université Lyon 1, INSA de Lyon, CNRS, UMR 5240, Villeurbanne, France. [3]Institut Pasteur, Université Paris Cité, CNRS UMR3525, Unité Plasticité du Génome Bactérien, Département Génomes et Génétique, Paris, France. [4]Namur Molecular Tech, UCLouvain, CHU UCL Namur, Yvoir, Belgium. [5]Université Paris-Saclay, CNRS UPR 2301, Institut de Chimie des Substances Naturelles, Gif-sur-Yvette, France. [6]National reference center for antimicrobial resistance in Gram negative, CHU UCL Namur, Yvoir, Belgium. ✉e-mail: sarah.bigot@cnrs.fr

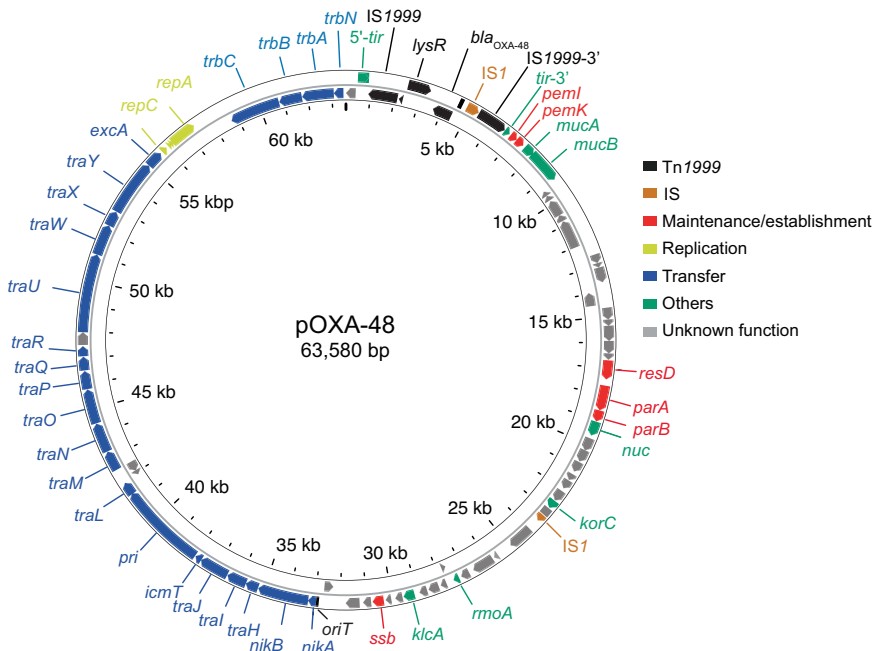

**Fig. 1 | Circular genetic map of the pOXA-48 plasmid.** The outer and inner arrows represent the coding DNA sequences (CDS) on the forward and reverse strands, respectively. Each CDS is color-coded according to its predicted function, as indicated in the figure legend. The categories include transposon elements (Tn*1999*), insertion sequences (IS), genes involved in plasmid maintenance/establishment, replication, transfer, and others, as well as genes with unknown function.

However, the factors and molecular mechanisms that facilitate the successful adaptation, maintenance, and spread of the pOXA-48 plasmid remain poorly understood. Conjugative plasmids often rely on functional elements for replication, stability, and conjugative transfer, which include replication and partitioning systems for stable inheritance, toxin-antitoxin systems that kill cells that have lost the plasmid, and conjugative machinery for horizontal gene transfer. Moreover, plasmid transfer imposes a fitness cost due to the synthesis of the type IV secretion system (T4SS), which drives the transfer of plasmid DNA into recipient cells. Plasmid-encoded regulatory factors are known to control the transfer and mitigate such fitness costs. While such factors have been yet identified for many conjugative plasmids of the IncF, IncX, IncA/C, and IncHI groups[8–14], none have yet been identified for pOXA-48. Understanding these mechanisms is crucial to determining how pOXA-48 balances the cost of transfer while ensuring its persistence across diverse bacterial hosts. While several genetic modules of pOXA-48 can be predicted based on its similarity to the IncM plasmid pCTX-M3[15], experimental validation has been scarce. To date, only the type II toxin-antitoxin system PemIK has been experimentally shown to contribute to plasmid maintenance and eliminate plasmid-free cells through a post-segregational killing mechanism[16]. Other factors involved in replication, stability, and conjugation remain largely unidentified experimentally.

In this study, we employed a transposon insertion sequencing (Tn-Seq) approach to systematically identify genes necessary for the maintenance, stability, and efficient transfer of the pOXA-48 plasmid in *E. coli*. By combining genetic, microscopy, bioinformatics, and structural modeling approaches, we have identified genetic determinants that ensure the success of pOXA-48 in bacterial populations.

## Results

### High-density transposon insertion library of the pOXA-48 plasmid

To validate the genetic elements required for pOXA-48 maintenance and transfer, we first sequenced the plasmid isolated from *Klebsiella pneumoniae*[1], revealing a 63,580 bp conjugative plasmid. The plasmid exhibits 95.6% nucleotide identity with the IncM pCTX-M3 plasmid[15], which allowed us to extrapolate its genetic content. Among the 83 predicted genes, several were annotated based on sequence similarity, including the *tra*, *trb* and *rep* genes, which are likely involved in conjugative transfer and replication, respectively (Fig. 1 and Table S1). Interestingly, 43% of the predicted genes remain uncharacterized.

To explore these unknown factors, we constructed a high-density transposon-insertion mutant library of the pOXA-48 plasmid in the *E. coli* K12 MG1655 strain, referred to as the input library. The Himar1 transposition cassette inserts either into the chromosome of *E. coli* or into the pOXA-48a plasmid. Two experimental replicates were generated, and the reads were aligned to the pOXA-48 plasmid. A total of 916,455 reads aligning to a TA dinucleotide border of pOXA-48 were mapped. Given that pOXA-48a contains 2788 TAs, the average insertion density was 52.3% and 50.9% for replicates 1 and 2, respectively, while the average number of reads per TA was 323.5 and 176.4 (Supplementary Data 1). For the analysis, given that the Pearson correlation between the two replicates was 0.76, we decided to aggregate the results from the two experimental replicates. This analysis identified 1671 unique insertion sites, with 59.9% of the 2788 TA dinucleotide sites of pOXA-48 showing successful insertion with the Himar-1 transposon, without any detectable insertion bias. To determine which genes are involved in plasmid transfer, we subjected the input library to two independent conjugation assays in which the mutated plasmid pool was transferred to an *E. coli* MG1655 recipient strain. Sequencing of the resulting output library 1 and 2 of transconjugants yielded a total of 28,844,013 reads, all of which mapped to the pOXA-48 plasmid, with a Pearson correlation between the two replicates of 0.91. Only 44.1% of the plasmid's TA sites showed insertions (Supplementary Data 1). These findings indicate that a substantial portion of the pOXA-48 plasmid plays a crucial role in the transfer process.

### Functional mapping of replication, segregation, and stability genes in the pOXA-48 plasmid

To identify the key elements required for pOXA-48 maintenance, we first analyzed the input transposon library. By calculating the Log1p of mean read counts for each gene, we identified seven genes with either null read values (*repA*, *repB*, *korC*) or low transposon insertion

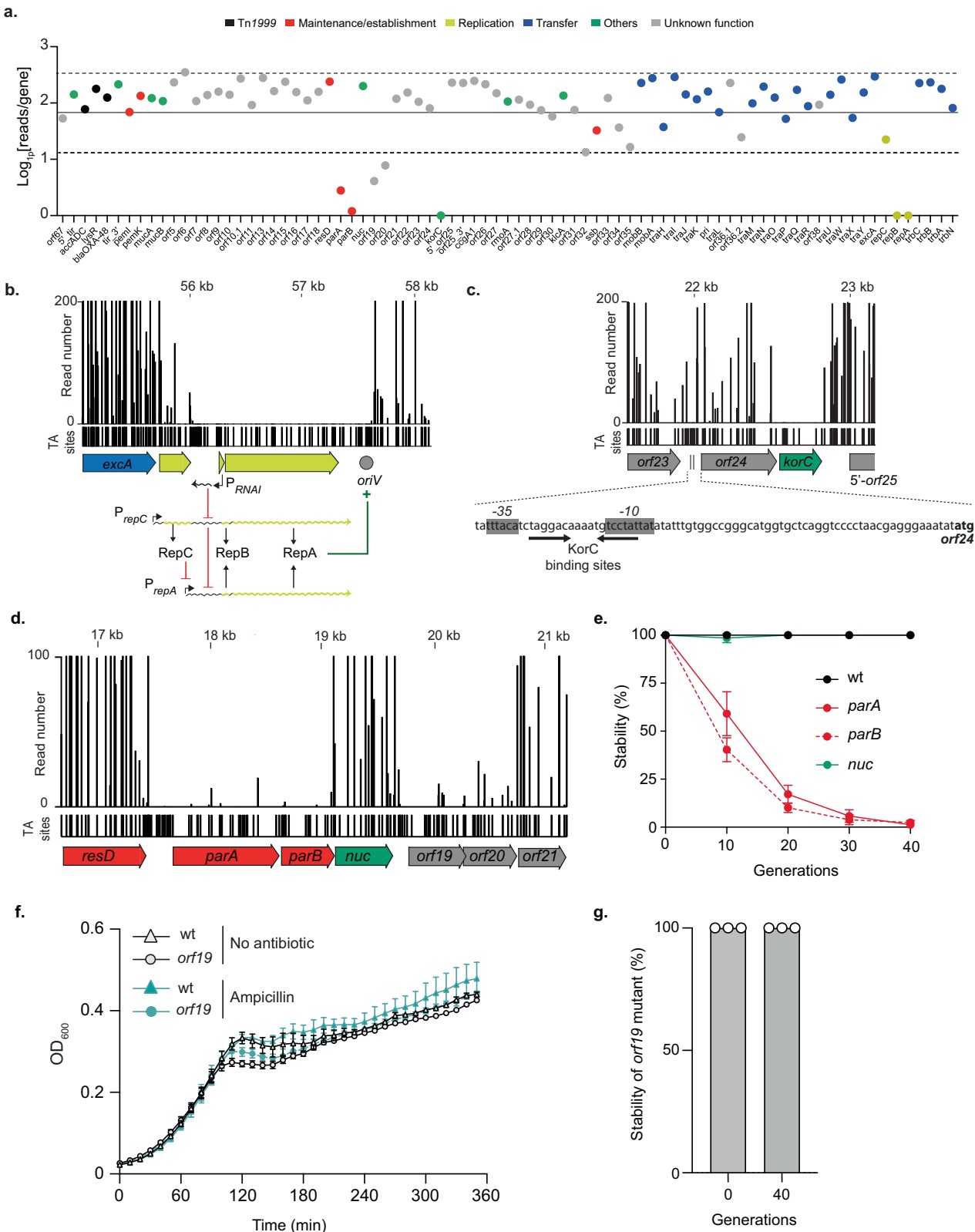

frequencies (*parA*, *parB*, *orf19* and *orf20*), indicating their potential roles in the maintenance of the pOXA-48 plasmid (Fig. 2a).

The replication of IncL/M plasmids relies on the RepA initiation protein, whose synthesis is translationally coupled to the expression of the leader peptide RepB. Plasmid copy number control occurs at two levels: transcriptionally, through the repressor protein RepC, which inhibits the activity of the $P_{repA}$ promoter, and post-transcriptionally,

via the antisense RNA molecule RNAI, which binds to its complementary sequence in the leader region of the *repA* mRNA, thereby inhibiting RepA translation[17,18] (Fig. 2b). The absence of transposon insertions in the region containing the predicted origin of replication (*oriV*) and the annotated replication initiation genes *repA*, *repB* underscores their essential role in the maintenance of the pOXA-48 plasmid (Fig. 2b).

**Fig. 2 | Identification of genes required for replication and maintenance of pOXA-48 plasmid. a** Tn-Seq analysis of transposon insertion abundance across the pOXA-48 plasmid represented as Log1p(reads/gene). The mean of reads (filled line) with SD (dotted lines) are represented. Genes are color-coded based on their predicted function. **b** Read coverage of transposon insertions around the *repCBA* locus. The schematic below shows the genomic context of these replication-related genes, highlighting regulatory elements, including the antisense RNA molecule RNAI and the origin of replication (*oriV*). **c** Tn-Seq read distribution for the *korC* gene and its neighboring region. The putative binding sites of KorC are shown below, along with the predicted promoter regions and the *orf24*'s codon start.

**d** Read counts for the transposon insertions across the *resD*, *parA*, *parB*, *nuc*, *orf19*, *orf20*, and *orf21* genes. **e** Stability assay of pOXA-48 plasmid derivatives with deletions of *parA*, *parB*, and *nuc* over 40 generations. The percentage of cells retaining the plasmid was measured at different time points, and the mean and SD of three independent clones are represented. **f** Effect of the *orf19* deletion on cell growth was monitored by measuring the optical density at 600 nm (OD$_{600}$). The mean and SD of three independent clones are shown. **g** Stability assay of pOXA-48 plasmid deleted of *orf19* at time 0 and after 40 generations. The percentage of cells retaining the plasmid is estimated from three independent clones (white dots).

Beyond the replication region, analysis of the average read counts showed a significant lack of transposon insertions in the *korC* gene, suggesting its potential role in plasmid maintenance (Fig. 2a, c). This hypothesis was further supported by a parallel study, which demonstrated that silencing *korC* expression by CRISPR interference led to a strong defect in pOXA-48 stability[19]. The *korC* gene encodes a putative transcriptional repressor that shares 39% homology with the KorC repressor of IncP RP4 plasmid, which is known to regulate the promoter regions upstream of *klcA*, *kleA*, and *kleC* genes, although its precise function in RP4 plasmid physiology remains undefined. A search for putative KorC binding sites from the IncL/M group identified a consensus sequence[20] which we found uniquely between the predicted −35 and −10 boxes of the *orf24-korC* operon promoter, suggesting potential autoregulation of *korC* expression (Fig. 2c). The absence of transposon insertions in this sequence suggests that the *korC* promoter is located here and supports the essential role of KorC in pOXA-48 maintenance.

The *parA* and *parB* genes, which showed lower insertion abundance (Fig. 2a, d), share 98% and 99% nucleotide identity, respectively, with the *parA* and *parB* genes of the IncM pCTX-M3 plasmid, which encode a partitioning system[21] (Table S1). Stability assays confirmed that the ParAB system of pOXA-48 plays a crucial role in the plasmid stable inheritance, as the deletion of either the *parA* or *parB* gene led to a rapid loss of the plasmid in the absence of selection pressure (Fig. 2e). Consistently, this loss was associated with a growth defect under selective pressure (Fig. S1). Interestingly, unlike the reported role of the putative nuclease Nuc in the partitioning of the pCTX-M3 plasmid[21], we found that Nuc of pOXA-48 does not contribute to plasmid partitioning, as its deletion did not affect plasmid stability (Fig. 2e).

We identified two genes of unknown function, *orf19* without any conserved domain, and *orf20* predicted to encode an XRE-like transcriptional regulator, both of which showed a lower abundance of transposon insertions in the pOXA-48 Tn-Seq library (Fig. 2a, d). This observation suggested that *orf19* and *orf20* might play essential roles in plasmid maintenance. To explore their potential contributions, we first generated a deletion of *orf19* and determined the effect of its absence on host cells. Deletion of *orf19* had no noticeable impact as the growth of pOXA-48 mutant and wild-type strains was similar in the presence or absence of selective pressure (Fig. 2f). Moreover, we evaluated the stability of the plasmid mutant and found that 100% of cells still contained the *orf19* plasmid mutant after 40 generations of growth (Fig. 2g). These findings suggested that the reduced number of transposon insertions observed for *orf19* in the Tn-Seq analysis may be attributable to a polar effect on the expression of *orf20* suggesting that *orf19-orf20* form an operon. In contrast, generating a plasmid deleted for *orf20* proved to be highly challenging, as its deletion was only obtained when *orf20* was provided in *trans* on a complementation plasmid, suggesting its essential role in plasmid maintenance. This finding is corroborated by the study of Calvo-Villamañán et al., which demonstrated that silencing *orf20* (DNDJGHEP_0014) for 24 h led to a drastic plasmid loss in *E. coli*[19].

To further explore the impact of acquiring an *orf20*-deleted plasmid, we performed real-time imaging using microfluidic chambers

in combination with the following fluorescence reporter system (Movie S1). We introduced a P$_{Tet}$-sfGFP fluorescent reporter, consisting of the *superfolder gfp* gene under the control of the P$_{Tet}$ promoter, into an intergenic region of the *orf20*-deleted pOXA-48 plasmid. The TetR-producing donor cells exhibited no fluorescence due to *sfgfp* gene repression by TetR and expressed *orf20* from the complementation plasmid pOrf20 to maintain the *orf20*-deleted plasmid. The TetR-free recipient cells carried the mCherry reporter system that conferred red fluorescence. Transconjugants acquiring the *orf20*-deleted plasmid relieve TetR repression and produced sfGFP, appearing yellow (red and green fluorescence) in overlay images (Movie S1). In these experiments, we initially mixed donor and recipient cells on a filter for 4 h prior to real-time imaging of the resulting conjugation mix. Microfluidic experiments were then performed with a constant flow of ampicillin, which selected for plasmid-maintaining cells that produced the carbapenemase OXA-48, enabling growth in the presence of ampicillin. We observed that recipient cells expressing *orf20* in *trans* acquired the plasmid, and the resulting yellow transconjugants proliferated under selection pressure. In contrast, the Orf20-free recipient cells did not successfully maintain the plasmid. Only a few transconjugants were observed, and these cells were elongated, indicating halted division despite their ability to resist ampicillin. Additionally, some red recipient cells exhibited abnormal elongation and increased cell width before undergoing bulging and eventual lysis, a phenotype characteristic of ampicillin-induced stress. These results suggest that these recipient cells initially acquired the *orf20*-deleted plasmid and likely lost it rapidly. The temporary production of the carbapenemase OXA-48 may have delayed ampicillin-induced killing, but the plasmid loss finally ultimately resulted in filamentation and then lysis. This highlights the crucial role of *orf20* in plasmid stability, explaining why a viable *orf20* deletion mutant could not be obtained.

## Identification and characterization of a novel type I toxin-antitoxin system

We identified an unannotated region between the *repA* and *trbC* genes (Fig. 1), which prompted further investigation into potential coding sequences within this region. Detailed analysis revealed a likely type I toxin-antitoxin (TA) system of a DqlB (*dinQ*-like B) family that was never experimentally validated[22]. This family consists of DqlB, a short helical hydrophobic toxic peptide, and the noncoding RNA *agrB* trans-acting antitoxin that should inhibit translation of *dqlB* by complementary base paring[22]. We identified a 35-nt sequence in the 5′ untranslated region (5′ UTR) of the *dqlB* mRNA that is complementary to the *agrB* RNA (Fig. 3a and Fig. S2a), suggesting that base-pairing between these elements could inhibit the translation of *dqlB*. Tn-Seq analysis supported the essentiality of this predicted antitoxin, as transposon insertions were not recovered in the predicted *agrB* RNA or its potential promoter region, indicating its crucial role in plasmid maintenance (Fig. 3a).

To confirm the functionality of this TA system, the *dqlB* gene, including its 5′ UTR, was cloned into a low-copy vector under the P$_{BAD}$ arabinose-inducible promoter. Production of the DqlB peptide, induced by arabinose, was found to be toxic, as no growth was observed under inducing conditions (Fig. 3b). Microscopic

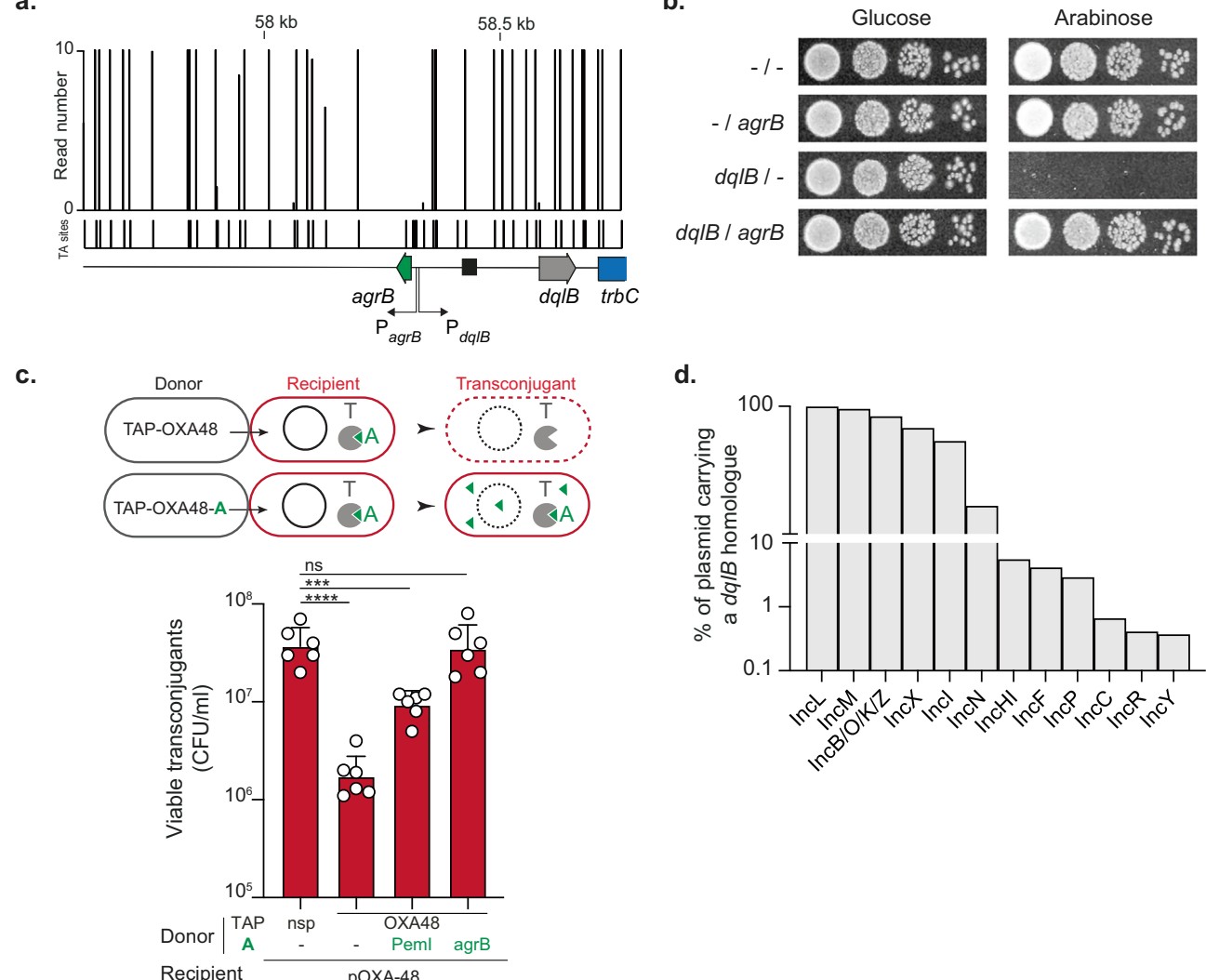

**Fig. 3 | Identification of a type I toxin-antitoxin encoded by the pOXA-48 plasmid. a** Tn-Seq data showing read counts for transposon insertions across the *agrB*/*dqlB* region of the pOXA-48 plasmid. The schematic below illustrates the genomic organization of the predicted toxin-antitoxin system, with *agrB* in green encoding the antitoxin (A) and *dqlB* in gray the toxin (T). **b** Spot assays evaluating the toxicity of DqlB and the protective effect of *agrB*. Growth of *E. coli* (−/−; LY4078), *E. coli* producing *agrB* (LY4080), DqlB (LY4079) or *agrB* and DqlB (LY4081) is shown under conditions with either glucose or arabinose. **c** Schematic representation of the Targeted Antibacterial Plasmid (TAP) system used to investigate the role of *agrB* and DqlB. Donor cells carry a TAP-OXA degrading the pOXA-48 plasmid, leading to cell death or a TAP-OXA-A targeting the pOXA-48 plasmid but producing the antitoxin gene to counteract the toxin activity and rescue the transconjugant cell. The histograms show the number of viable transconjugants (CFU/mL) after acquisition of the different TAPs. Donors TAP-nsp (LY1369), TAP-OXA48 (LY1522), TAP-OXA48-PemI (LY1549), TAP-OXA48-agrB (LY4038); Recipients (MS388). Mean and SD are calculated from six independent experiments. *P*-value significance from Tukey's multiple comparisons statistical analysis is indicated by ns: not significant, *** (*p* = 0.0002), **** (*p* < 0.0001). **d** Histograms showing the proportion of Inc plasmids containing a *dqlB*-like homolog.

examination of DqlB-producing cells revealed the presence of ghost cells (Fig. S2b), suggesting that DqlB targets the inner membrane and compromises its integrity through a mechanism similar to its DinQ homolog[23] or the Hok toxin from plasmid R1[24,25]. Furthermore, co-expression of *agrB* in *trans* provided protection against the toxic effects of DqlB, confirming that *agrB* functions as an antitoxin (Fig. 3b). To determine whether repression of *dqlB* by *agrB* relies on a complementary base-pairing mechanism between the *agrB* RNA and the 5′ UTR of *dqlB*, we constructed an *agrB* mutant with a 4-nt mismatch in the predicted pairing region (Fig. S2a). This mutant failed to repress the toxicity induced by wild-type DqlB (Fig. S2c). We then generated a compensatory mutation into the 5′ UTR of *dqlB*, restoring complementarity to the mutated *agrB*. Reestablishing base pairing fully restored the ability of the *agrB* mutant to neutralize DqlB toxicity

(Fig. S2c). These results demonstrate that specific base-pairing interactions are critical for *agrB* to function as an effective antitoxin.

To validate DqlB-*agrB* as an addiction module, *i.e.* a system that promotes post-segregational killing of plasmid-cured cells, we took advantage of an antibacterial strategy previously designed to cure the pOXA-48 plasmid[16]. In this system, a mobilizable Targeted Antibacterial Plasmid (TAP) encoding a CRISPR system targeting the pOXA-48 plasmid (TAP-OXA48) is transferred by conjugation in pOXA-48-containing cells. CRISPR-mediated cleavage of pOXA-48 leads to its loss, which triggers post-segregational killing though antitoxin depletion, toxin activation and cell death (Fig. 3c). Providing antitoxins by cloning them on TAP (TAP-OXA48-A) prevent this viability loss by preventing toxin activation (Fig. 3c). This approach was previously used to validate the activity of the PemIK type II TA system encoded on

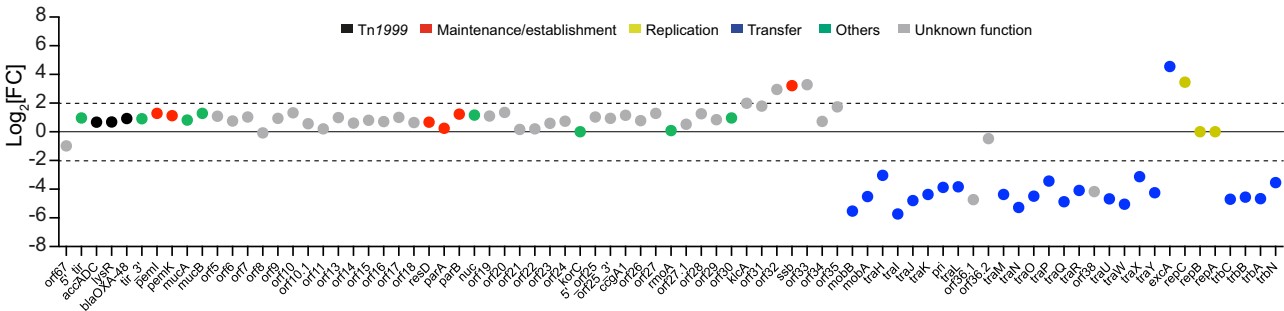

**Fig. 4 | Identification of genes essential for conjugation and enriched in transposon insertion sites.** $Log_2(FC)$ analysis of transposon insertion abundance comparing the output library to the input library. Threshold of 2 and −2 are indicated by the dotted lines. Genes are color-coded based on their predicted function.

the pOXA-48 plasmid. In that case, production of the PemI antitoxin by a TAP-OXA48-PemI resulted in partial rescue of transconjugant viability compared to transconjugants acquiring a non-specific TAP (TAP-nsp) which did not degrade the plasmid and thus did not activate any TA system[16] (Fig. 3c). This partial rescue suggested that another yet unidentified TA system could be responsible for the remaining viability loss that could not be complemented through PemI production. We therefore assess the role of DqlB-*agrB* in this loss of viability by cloning *agrB* in TAP-OXA48. Remarkably, expression of *agrB* alone was sufficient to rescue killing by TAP-OXA48, confirming that DqlB-*agrB* induces cell death upon plasmid loss and therefore functions as an addiction module (Fig. 3c).

Given that DqlB-*agrB* is a major stability determinant of pOXA-48, we investigated the conservation of this system across various plasmids. To address this question, we mined the PLSDB database[26,27] for *dqlB* homolog, using an Hmm model based on seven known *dqlB*[22] homologs and the pOXA-48-encoded *dqlB* nucleotide sequence as a query. A total of 3310 plasmids carrying a copy of this gene was found among a subset of 15775 plasmids for which the incompatibility group was well defined (Fig. S3a). Homologs of *dqlB* are ubiquitous in the IncL/M taxonomic unit, with 99.3% of IncL and 95.5% of IncM plasmids encoding a copy of *dqlB*. The *dqlB* gene is also prevalent on IncB/O/K/Z (83.2%), IncI1 (67.4%) and IncX (53.3%) (Fig. 3d). Homologs can also be found uncommonly among IncN (16.6%), IncHI (5.5%), IncF (4.2%) and IncP1 (2.9%), and sporadically on IncC (0.66%), IncR (0.4%) and IncY (0.37%) plasmids. Notably, the 3310 *dqlB* homologs identified during our screen were highly redundant. To study the diversity of these homologs, identical sequences were collapsed together and used to build a maximum likelihood tree. Each leave was ultimately decorated with the incompatibility group distribution of their plasmids (Fig. S3b). Overall, *dqlB* homologs are distributed in clusters corresponding to the incompatibility group of their vector, with the exception of IncF-encoded homologs, which are found scattered across the tree, suggesting that IncF plasmids are prone to acquire this TA system by horizontal gene transfer from another plasmid. Interestingly, *dqlB* homologs from pOXA-48 and other IncL/M plasmids form a monophyletic group with IncN-encoded homologs (Fig. S3b), which could therefore indicate a common origin for the acquisition of this toxin between IncL/M and IncN plasmids. Overall, these results show that the DqlB-*agrB* TA system can be commonly found among plasmids. In practice, however, its presence is currently likely overlooked as it is not always annotated accordingly due to its small size.

## Genes required for efficient pOXA-48 plasmid transfer

The comparison between the initial transposon library and the post-transfer library provided insights into the genes required for efficient plasmid transfer. The analysis of the $Log_2FC$ (fold-change) indicated that 24 genes were underrepresented in the transconjugant population, suggesting their essential role in the conjugative transfer process (Fig. 4). Specifically, disruptions in *tra* and *trb* genes encoding for the

T4SS components (Table S1) led to a marked reduction of transposon insertion reads, thus confirming their key role in the transfer of the pOXA-48 plasmid. Interestingly, we also identified two hypothetical proteins, Orf36.1 and Orf38, with unknown functions, whose disruption appeared to significantly impact transfer efficiency, suggesting that these proteins could represent novel factors involved in plasmid conjugation (Fig. 4). To further investigate the roles of *orf36.1* and *orf38* in conjugation and validate the Tn-Seq findings, we assessed the impact of deleting these genes on conjugation efficiency (Fig. 5). Deletion of *orf38* led to a $10^6$-fold decrease in conjugation efficiency, whereas deletion of *orf36.1* caused only a threefold reduction (Fig. 5), indicating that Orf38 is more critical for conjugation than Orf36.1. The ectopic production of Orf36.1 from a complementation plasmid fully restored conjugation frequency to the wild-type levels, and production of Orf38 in the *orf38* mutant led to a $10^5$-fold increase in conjugation frequency. However, this complementation remained partial, as it did not fully restore the transfer frequency to that of the wild-type.

Orf36.1 is a small protein consisting of 65 amino acids and contains a conserved domain from the Hha superfamily (Table S1), which is found in proteins involved in modulating of bacterial gene expression, and can influence processes such as bacterial virulence, horizontal gene transfer and cell physiology[28]. Although the precise role of Orf36.1 in the pOXA-48 transfer process remains unknown and further investigations are needed, we hypothesize that it may function as a transcriptional regulator, promoting plasmid transfer by modulating the expression of genes essential for the pOXA-48 conjugation.

To validate the role of Orf38 in plasmid transfer, we conducted an in silico analysis to better understand its potential function. Orf38 is a 164 amino acid protein without any identifiable conserved domains (Table S1). In silico analysis did not predict the presence of a signal peptide or transmembrane domain, suggesting that Orf38 is likely localized in the cytoplasm. To gain further insights into its function, we used AlphaFold 3.0 to generate a structural model of Orf38, which showed an average predicted local distance difference test (pLDDT) score of $79.6 \pm 12.6\%$. A subsequent search for homologous structures in the Protein Data Bank using DALI[29] identified DotN, a component of the T4SS in *Legionella pneumophila*[30], as the best and only matching structure spanning the entire length of Orf38 model. Our model of Orf38 shared a root mean square deviation (RMSD) of 5.4 with the crystal structure of DotN (PDB 5X1H), indicating significant structural homology (Fig. S4a). Although the precise function of DotN remains uncertain, it is known to form part of a multimeric complex with the DotM and DotL T4SS components, forming the Type IV coupling complex (T4CC), which plays a role in recognizing secreted effectors[31]. To further explore whether Orf38 might be part of a similar complex, we modeled its interaction with proteins homologous to DotL and DotM, which correspond to TrbC and TrbA, respectively, in the pOXA-48 plasmid (Table S1 and Fig. S4b). The resulting structure showed strong similarity to the cryo-electron microscopy (cryo-EM) structure of the T4CC from *L. pneumophila* (PDB 6SZ9), with an RMSD of 2.8 Å,

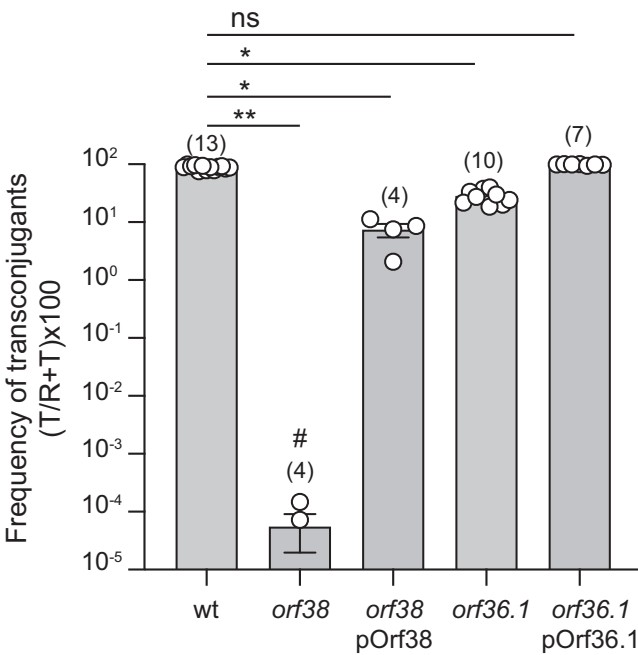

**Fig. 5 | Impact of *orf38* and *orf36*.1 deletions on conjugation frequency.** Histograms showing the frequency of transfer after 4 h of mating for the wild-type (wt), *orf38* and *orf36*.1 deletion mutants, as well as their complementation with pOrf38 and pOrf36.1 plasmids producing respectively Orf38 and Orf36.1. Data are represented as mean and SD from (n) independent experiments (white circles). *p*-value significance from Kruskal-Wallis test corrected with Dunn's multiple comparisons test is indicated by ns (not significant), * ($p = 0.023$, pOrf38; $p = 0.0461$, *orf36.1*), ** ($p = 0.0013$). # indicates two replicates with a detection limit of transconjugants below $10^{-8}$. Donors LY1844, LY3303, LY3697, LY3739, LY3747; Recipient LY945.

suggesting that the T4SS of pOXA-48 may contain a T4CC closely related to that of *L. pneumophila*. However, the cryo-EM structure did not include the N-terminal regions of DotM and DotL, which were reported as disordered transmembrane regions[31]. In contrast, our AlphaFold model predicted these N-terminal regions for TrbA and TrbC, the homologs of DotM and DotL in pOXA-48. We subsequently generated a complete model of the DotN/DotM/DotL complex using AlphaFold, as shown in the right panel of Fig. S4b, which includes these previously unstructured N-terminal regions. The presence of these regions in our model further supports the similarity between the Orf38-TrbA-TrbC and DotN-DotM-DotL complexes. While significant similarities exist between these complexes, key differences were also noted. Notably, TrbC appears to lack the swinging C-terminal tail of DotL after the α-helix16 region, which was shown to be essential for the secretion of a subset of effectors in *L. pneumophila*[31]. This absence could reflect functional divergence between the T4SS of pOXA-48, which is primarily involved in DNA transfer, and the T4SS of *L. pneumophila*, which is dedicated to secrete over 300 effectors.

**Analysis of genes with enrichment in transposon insertions**
Transposon insertion can disrupt genes that function as repressors of conjugation, potentially enhancing plasmid transfer efficiency. In our Tn-Seq analysis, we observed an enrichment of transposon insertions in five genes: *orf32*, *ssb*, *orf33*, *excA*, and *repC*, with a Log₂FC above the defined threshold (Fig. 4 and Fig. S5). This suggested that disrupting these genes might increase conjugation efficiency. To test this hypothesis, we constructed deletion mutants for each of these genes and assessed their impact on conjugation efficiency compared to the wild-type plasmid. Surprisingly, conjugation experiments showed that only the deletion of *orf32* and *repC* significantly affected the frequency of plasmid transfer, with conjugation rates respectively 1.87-fold and

1.56-fold higher than those observed for the wild-type (Fig. 6a). The ectopic production of RepC and Orf32 from complementation plasmids restore the efficiency of conjugation to the wild-type level.

We hypothesized that the increased transposon insertions could also be linked to an alteration in plasmid copy number (PCN). To explore this possibility, we measured the PCN of each deletion mutant and compared it to the wild-type plasmid. Our analysis revealed that only the *repC* deletion mutant exhibited a significant increase in PCN, with a 1.9-fold higher copy number compared to the wild-type, and that ectopic production of RepC from a complementation plasmid restores the PCN to a wild-type level (Fig. 6b). This increase in PCN was also associated with heightened resistance to imipenem, likely due to the higher number of copies of the $bla_{OXA-48}$ gene, as previously reported[5] (Fig. 6c). These results confirm the predicted role of RepC as a regulator of PCN, certainly by repressing the promoter activity of *repA*, and associated with the increase transfer efficiency help explain the reason for the increased number of reads observed in the Tn-Seq results.

For the remaining genes - *orf32*, *ssb*, *orf33*, and *excA* - the reason behind the enrichment in Tn-Seq reads is less clear. Bioinformatic analysis suggests that *ssb* and *orf33* form an operon with *orf32*, and their disruption should not affect the expression of the upstream *orf32* gene. Both *orf32* and *orf33* lack conserved domains, leaving their function unknown (Table S1). The *ssb* gene encodes a single-stranded DNA-binding protein that binds single-stranded DNA. While the precise role of plasmid-encoded Ssb remains unclear, studies on the F plasmid suggest that Ssb facilitates the initial round of plasmid duplication in newly formed transconjugant cells[32]. Additional studies are required to elucidate the specific roles of these three genes in the process of pOXA-48 plasmid transfer. The *excA* gene is predicted to function as an exclusion gene, potentially preventing the acquisition of additional plasmids by host cells that already harbor the pOXA-48 plasmid. To validate this hypothesis, we assessed the efficiency of exclusion in an *excA* deletion mutant by calculating the exclusion index (EI)[33]. The EI was determined by comparing the frequency of plasmid acquisition in a recipient strain containing the pOXA-48 plasmid to that of an isogenic plasmid-free strain (Fig. 6d). For the wild-type pOXA-48 plasmid, the EI was approximately 490, indicating that *E. coli* cells harboring pOXA-48 are about 490 times less likely to acquire an additional plasmid compared to plasmid-free cells. In the *excA* deletion mutant, the EI dropped significantly to a value of 2, suggesting that the exclusion mechanism was effectively abolished in the absence of *excA* (Fig. 6d). To confirm the role of *excA* in this exclusion activity, we introduced a complementation plasmid encoding *excA* into the deletion mutant. The ectopic production of ExcA restored the exclusion phenotype, demonstrating that ExcA is essential for the exclusion function of the pOXA-48 plasmid. These results established ExcA as a key component of the exclusion mechanism, though further investigation is needed to explain the increased number of reads for this gene in the Tn-Seq data, given the observation that approximately 50% of total reads aligned with the *excA* gene (Fig. S5). The lack of PCN increase in *excA* mutants is surprising, given that clonal populations of these mutants constantly acquire additional plasmids, which should theoretically result in a higher plasmid number per cell. Like most conjugative plasmids, pOXA-48 conjugates at a very low frequency in liquid media but at a much higher frequency on solid surfaces. We hypothesized that *excA* mutants grown in liquid media conjugate at low frequency in this condition, leading to the observed low PCN estimated from stationary phase liquid cultures (Fig. 6b), while colonies grown on solid agar plates would exhibit continual transfer of the *excA* mutant plasmid, resulting in a significantly higher PCN per cell compared to liquid culture. To test this, we measured the PCN of bacteria directly collected from agar plates. The results confirmed that *excA* mutants collected from plates had a PCN nearly 10-fold higher than that of the wild type (Fig. 6e). Furthermore, ectopic production of

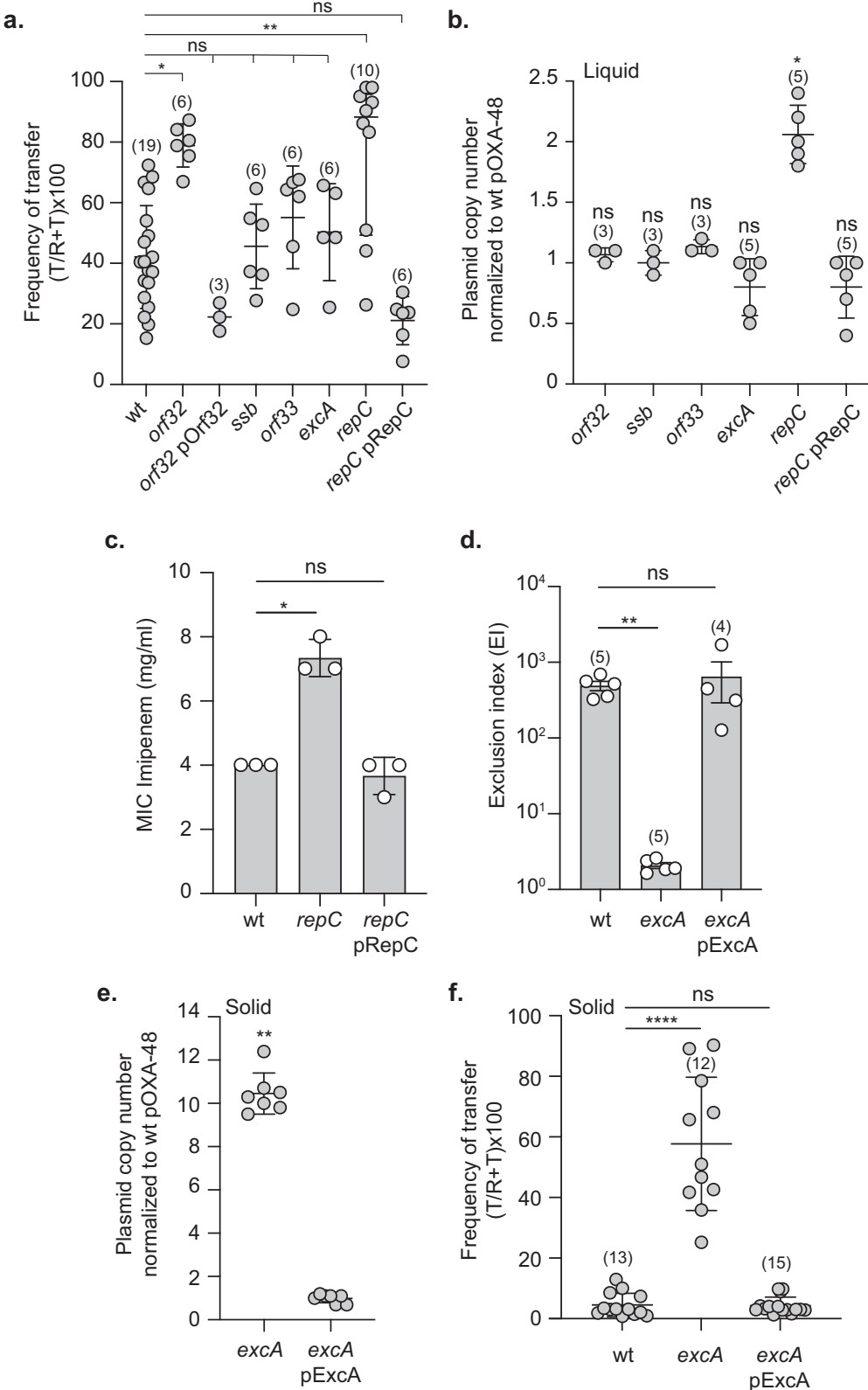

ExcA restored the PCN to wild-type levels. To further investigate the biological significance of this PCN increase, we assessed the conjugation frequency of the *excA* mutant when donor cells were collected directly from colonies grown on agar plates. In this condition, the *excA* mutant displayed a 12-fold increase in conjugation frequency compared to the wild type, and this phenotype was reversed by ectopic expression of *excA*, restoring conjugation to wild-type levels (Fig. 6f).

These results demonstrate a direct correlation between elevated PCN and enhanced conjugation frequency, consistent with observations made for the *repC* mutant. Most importantly, they provide an explanation for the enrichment of transposon insertions in *excA* observed in the Tn-Seq output library. Since donors for conjugation in the Tn-Seq experiment were prepared from colonies grown on solid medium, the elevated PCN and conjugation rate of *excA* mutants likely conferred a

**Fig. 6 | Analysis of phenotypes associated with deletion of *orf32*, *ssb*, *orf33*, *excA*, or *repC* genes. a** Conjugative transfer frequency after 150 min of mating for the wild-type (wt) (LY1844) and mutants, *orf32* (LY4148) and complemented strain with pOrf32 plasmid (LY4192), *ssb* (LY4149), *orf33* (LY4153), *excA* (LY3260), and *repC* (LY3259) and complemented strain with pRepC plasmid (LY3606). Recipient LY945. Data are represented as mean and SD from at least six independent experiments (gray circles). Comparisons test is indicated by ns (not significant), *(*p* = 0.0133), **(*p* = 0.0095). **b** Plasmid copy number (PCN) for each mutant and *repC* complemented strain (*repC* pRepC) from stationary phase culture. The ratio of the mutant PCN to the wild-type PCN is represented. Data are represented as mean and SD from (n) independent experiments (gray circles). Comparisons test is indicated by ns (not significant), *(*p* = 0.044) **c** Minimum inhibitory concentration (MIC) of imipenem inhibiting the growth of the wild-type (wt), *repC* mutant, and *repC* complementated strain (*repC* pRepC). Data are represented as mean and SD from three independent experiments. Comparisons test is indicated by ns (not significant), *(*p* = 0.0463). **d** Histogram showing the exclusion levels of wild-type pOXA-48 transfer from donor to the recipients cells carrying wild-type pOXA-48 (wt), *excA* mutant with or without pExcA complementation plasmid. Data are represented as the mean and SD from (n) independent experiments (white circles). Comparisons test is indicated by ns (not significant), **(*p* = 0.0081). **e** PCN for *excA* mutant and complemented strain (*excA* pExcA) derived from colonies scratched from LB agar plates (solid). The ratio of the mutant PCN to the wild-type PCN are represented. Data are represented as mean and SD from seven independent experiments (gray circles). Comparison test is indicated by ns (not significant), **(*p* = 0.0011). **f** Conjugative transfer frequency after 60 min of mating for the wild-type (wt) and *excA* mutant, and complemented strain with pExcA derived from colonies scratched from LB agar plates (solid). Data are represented as mean and SD from (n) independent experiments (gray circles). Comparisons test is indicated by ns (not significant), ****(*p* < 0.0001). **a–e** *p*-value significance from the Kruskal-Wallis test corrected with Dunn's multiple comparisons test.

strong selective advantage during the transfer process, leading to their overrepresentation in the transconjugant population.

## Discussion

The transposon insertional analysis and functional assays presented in this study provide an integrated view of the genetic elements involved in the biology of the pOXA-48 plasmid. We identified multiple factors contributing to plasmid stability, including a novel toxin-antitoxin system, components of the replication machinery, elements of the type IV secretion system, and regulatory factors critical for conjugation.

Although the IncM pCTX-M3 and IncL pOXA-48 plasmids share 95.6% nucleotide identity, they differ notably in their replication and exclusion systems, placing them in different incompatibility groups[34]. The replication genes are highly conserved (Table S1), but differentiation occurs primarily based on the identity of their RNAI sequences, which show significant divergence. The replicon of pOXA-48, composed of RepC, RNAI, RepB, and RepA, closely resembles the components of the R1 plasmid replicon from the IncFII group, which includes CopB, the antisense RNA *copA*, Tap and RepA[35]. Our data demonstrate that RepC regulates the plasmid copy number (PCN), as its disruption or deletion leads to increased PCN. Notably, this elevated PCN in *repC*-deleted mutants correlates with enhanced conjugation rates, resembling the observations made for the R1 plasmid, where increased PCN is also linked to higher conjugation rates[36]. Interestingly, a recent study on pOXA-48 described the isolation of an evolved plasmid variant from a clinical patient sample, which exhibited a higher PCN associated with increased antimicrobial resistance (AMR)[5]. However, in that study, no corresponding increase in conjugation frequency was observed. This discrepancy may be explained by the location of the mutation, which was identified upstream of the *repA* gene and did not affect *repC*. These findings suggest that while mutations impacting *repC* can simultaneously influence PCN and conjugation frequency, mutations outside this region may not exert the same regulatory effect on transfer efficiency.

Further differences between the pOXA-48 and pCTX-M3 plasmids are observed in their exclusion systems. The *traX*, *traY*, and *excA* genes of the pOXA-48 and pCTX-M3 plasmids are more divergent from one another than other genes, encoding proteins with significantly lower amino acid identity (Table S1). While the exact origin of this divergence remains unknown, one hypothesis is that a recombination event occurred during the evolutionary history of these plasmids. It has been proposed that both pOXA-48 and pCTX-M3 were derived from the ancestral pEL60 plasmid[37]. Interestingly, pCTX-M3 harbors *traX*, *traY*, and *excA* genes that are nearly identical to those of pEL60, whereas the corresponding genes in pOXA-48 have substantially diverged. This suggests that a gene exchange event may have introduced a variant region in the pOXA-48 lineage, replacing the original gene cluster found in the ancestral plasmid. In this study, we identified ExcA as the

entry exclusion protein for the pOXA-48 plasmid. Although the precise mechanism of plasmid exclusion remains unclear, exclusion in IncI plasmids is driven by the recognition of TraY by ExcA[38], similar to the TraG protein of plasmid F, which interacts with the exclusion protein TraS[39]. Interestingly, the region of TraY recognized by ExcA varies between plasmids; for instance, the central region is targeted in plasmid R64, whereas the C-terminal region is recognized in plasmid R621a[38]. The divergence in the *excA* region between pCTX-M3 and pOXA-48 thus reflects the specificity of exclusion mechanisms, and these plasmids do not mutually exclude one another[37]. The role of *traX* in this exclusion process remains unknown, but it is noteworthy that *traX* is part of the recombined region and appears genetically linked to *traY* and *excA*. While initial conjugation assays performed in liquid medium showed no impact of *excA* deletion on pOXA-48 transfer, experiments using donor cells from solid medium revealed a high increase in transfer frequency. This contrasts with the pCTX-M3 plasmid, where *excA* deletion has been shown to affect plasmid transfer[40].

Importantly, our study reveals additional factors involved in plasmid maintenance and stability. Beyond the well-characterized Par partitioning system, which ensures stable inheritance of the plasmid, we identified two other genes, *korC* and *orf20*, whose deletion was detrimental to cell growth under selective pressure, as well as a novel type I toxin-antitoxin system. The regulatory role of KorC in pOXA-48 appears similar to that in the RP4 plasmid, where the absence of Kor functions leads to a host-lethal phenotype[41]. In RP4, KorC represses the *klcA* gene, which encodes a putative anti-restriction protein, with a KorC binding site found in the *klcA* promoter. Interestingly, pOXA-48 contains a *klcA* homolog downstream of *korC*, near the origin of transfer (Fig. 1). However, no KorC binding site was predicted upstream of this gene, and only one site was found, located in the predicted *korC* promoter upstream of *orf24*, which may be in the same operon as *korC* (Fig. 2c). This suggests that KorC may autoregulate its own expression, as seen in other KorC systems. Further experiments are needed to determine the role of KorC in pOXA-48 and whether it regulates genes with toxic potential or modulates stress responses, thereby contributing to plasmid stability and host viability. Similarly, Orf20 also plays a critical role in plasmid maintenance. Our findings show that a plasmid deleted of *orf20* could be maintained when *orf20* was provided in *trans*, indicating an essential role in plasmid stability. This phenotype resembles that observed in the absence of KorC, which results in plasmid loss. Sequence analysis predicts that Orf20 is an XRE-like transcriptional regulator, suggesting a potential regulatory function similar to KorC. Further studies are required to elucidate the precise function of Orf20 in pOXA-48 physiology.

The essentiality of *orf20* and *korC* was also reported by Calvo-Villamañán et al., who used a plasmid-wide CRISPRi screen to investigate pOXA-48 fitness effects in various clinical strains of enterobacteria[19]. While our experiments were conducted in the laboratory strain *E. coli* K12 MG1655, their study included both clinical

isolates and the laboratory strain *E. coli* J53. They showed that silencing *pemI*, *pri* and *H-NS* (*orf36*.2) in J53 had minimal effects, consistent with our findings in MG1655, where no significant decrease in transposon insertions was observed for these genes. In contrast, expression of these genes is essential in clinical isolates, highlighting the importance of plasmid-host interactions and supporting the idea that gene essentiality is context-dependent, influenced by both genetic and physiological characteristics of the host strain.

In this study, we also identified several genes critical for pOXA-48 transfer, including components of the relaxosome and T4SS. However, the exact roles of some components, such as TraL, TraQ, TraR, TraW, TraX, and TrbB (Table S1), remain to be fully elucidated. Interestingly, we identified Orf38, previously uncharacterized as a T4SS component, as an essential factor for plasmid transfer. Structural modeling suggests that Orf38 forms a type IV coupling complex (T4CC) with the T4CP TrbC and TrbA. Such a complex is absent in "minimized" T4SS-A systems, like those found in R388 or pKM101 plasmids, but is present in "expanded" T4SS-B systems, such as the one in *L. pneumophila*[42]. In this bacteria, the T4CC acts as a platform for recruiting and secreting effectors. The role of such a T4CC in conjugative T4SS dedicated to DNA transfer remains unclear. While it is known that, in addition to DNA, proteins such as relaxase can be transferred through T4SS by covalent attachment to the DNA, other plasmid-encoded proteins may also be co-transferred in a DNA-dependent manner[43], or independently, as seen with the primase Sog of the IncI ColIb plasmid[44].

In recent years, increasing attention has been given to clinically relevant conjugative plasmids from groups such as IncI1[45,46], IncA/C[47], and more recently, IncX[48], which are implicated in the dissemination of resistance to expanded-spectrum beta-lactams, carbapenems, and colistin. These studies have uncovered numerous molecular factors and highlighted that the biology of conjugative plasmids remains an extensive field for exploration. Our work contributes significantly to advancing our understanding of the molecular mechanisms underlying pOXA-48 transfer, identifying critical genetic elements that warrant further investigation.

## Methods

### Bacterial strains, plasmids and growth

Bacterial strains are listed in Table S2, plasmids in Table S3, and oligonucleotides in Table S4. Gene deletion on the pOXA-48 plasmid used λRed recombination[49,50]. Modified pOXA-48 plasmids were transferred to the background strain *E. coli* K12 MG1655 by conjugation. The kan or Cm resistance cassette were removed using the site-specific recombination of the Flp recombinase from pCP20 plasmid[49]. Plasmid cloning was performed by Gibson Assembly[51]. Strains and plasmids were verified by Sanger sequencing (Eurofins Genomics and Microsynth).

Cells were grown at 37 °C in Luria-Bertani (LB) or Rich Defined Medium (RDM) broth. When appropriate, supplements were used in the following concentrations; Ampicillin (Ap) 100 µg/ml, Chloramphenicol (Cm) 20 µg/ml, Kanamycin (Kan) 50 µg/ml, Streptomycin (St) 20 µg/ml, Erythromycin (Erm) 100 µg/ml, Tetracycline (Tc) 10 µg/ml and diaminopimelate (DAP) 300 µM.

### pOXA-48 genome sequencing

Overnight cultures in LB medium of LY1844 strain was pelleted at 15000 g for 5 min. Supernatant was discarded, and pellet was resuspended in 525 µl of lysis buffer (CleanNA, Waddinxveen, The Netherlands) for bacterial DNA extraction. Semi-automated DNA extraction on a KingFisher platform (Thermo Fisher Scientific, Waltham, MA, USA) was performed with the Clean Pathogen kit (Clean NA) according to manufacturer's recommendations. Briefly, a mix of 73 µl of proteinase K and proteinase K solution were added to each sample prior vortexing and incubation for 15 min at 70 °C. Samples were centrifuged at 17,500 g for 10 min and 300 µl of supernatant were added to

620 µl of beads mix in a deep well plate. Two plates of Wash Buffer 1, 1 plate of Wash Buffer 2 and 1 plate of elution buffer were prepared. All plates were loaded on the KingFisher device and the extraction program was started. Eluted DNA (100 µl) were stored at −20 °C until use or directly processed to sequencing.

Bacterial DNA was sequenced using the Rapid Barcoding Kit SQK_RBK114.96, FLO-MIN114 flow cells, and a GridION (Oxford Nanopore Technologies, OX4 4DQ, UK) following the manufacturer's procedure. Four hundred ng of QC bacterial DNA were added to nuclease-free water to reach a volume of 20 µl per sample. Two µl of individual barcode were added to each sample before incubation at 30 °C for 2 min and 80 °C for 2 min. The pooled DNA library was purified with AMPure beads at a 2:1 DNA: beads ratio. One µl of Rapid Adapter solution was added to 800 ng of library to reach a maximum volume of 12 µl before incubation at RT for 5 min. of Sequencing Buffer (37.5 µl) and of Library Beads (25.5 µl) were added to the DNA library, and the mix was loaded on the flow cell and sequenced in "super accurate" mode. Generated data were basecalled with Dorado 7.3.11 and analyzed with in house bioinformatics pipeline. Data are available at Bio-Project PRJEB88520.

### High-density himar-1 transposon mutagenesis

Overnight cultures in LB medium of donor *E. coli* MFD*pir*+ carrying pSAM_Ec and recipient *E. coli* MG1655 carrying pOXA-48 were mixed at a ratio 1:1 in presence of 300 mM DAP, and incubated on filters (MF Membrane Filters-0.45 µm HA-Ref HAWP02500 Millipore) placed on LB-agar plates 4 h at 37 °C. The mating mixture was then resuspended in LB medium and spread onto LB-agar plates supplemented with Kan and Ap to select clones carrying the Himar-1 transposon and the pOXA-48 plasmid, respectively. After overnight incubation at 37 °C, about $10^6$ resistant colonies, designated as the input library, were collected, resuspended in LB medium and frozen in 40% glycerol at −80 °C. 1 ml of the frozen input library was washed and used as the donor in a mating of ratio 1:1 with an *E. coli* MG1655 recipient strain. The mating mix was spotted on filters, incubated for 4 h at 37 °C, then resuspended LB medium and entirely spread onto LB-agar plates supplemented with Kan and Cm to select transferred plasmids carrying transposon. After overnight incubation at 37 °C, approximately 2,000,000 transconjugant colonies were collected, providing an estimated ~715-fold theoretical coverage of the 2788 potential TA insertion sites in the pOXA-48 plasmid. This ensured robust representation of the mutant library in the output pool, despite the relatively low conjugation frequency (~0.3%). The resulting output library was resuspended in LB medium and frozen in 40% glycerol at −80 °C.

### Preparation of Tn-Seq sequencing library

Library DNA preparation and sequencing procedures were adapted from Royet et al.[52]. Genomic DNA (gDNA) from the mutant library was extracted using the Promega Wizard Genomic DNA Purification kit. Then, 30 µg of gDNA was digested with 30 U of MmeI in a final volume of 1.2 ml at 37 °C for 1.5 h. Digestions were heat-inactivated for 20 min at 65 °C and purified with a ratio of 0.45X of the magnetic beads Mag-Bind Total Pure NGS (Omega Biotek), allowing the removal of fragments smaller than 1000 bp. 2 µg of digested and purified DNA was ligated to double-stranded barcoded adapters at 0.55 µM. The ligation was performed at 16 °C overnight using 800 U of T4 DNA ligase in a final volume of 40 µL. Ligated DNA were purified using a ratio of 0.6X of the magnetic beads Mag-Bind Total Pure NGS (Omega Biotek) to eliminate the fragments smaller than 500 bp. To amplify the DNA adjacent to the transposon, a total of six 18-cycle PCRs were conducted for each sample. Each of the PCR mixtures comprised 40 ng of DNA, 1 U of Q5 DNA polymerase (New England Biolabs), 1X Q5 buffer, 0.2 mM of dNTPs, and 0.4 µM of the L1204-I1 and L1204-I2 primers. The six PCR products per sample were then combined, concentrated to a volume of 50 µl using a vacuum concentrator and loaded onto a 2% (w/v)

agarose gel. The 125 bp bands were excised and purified using a QIA-quick Gel Extraction kit (Qiagen). All gel electrophoresis and gel extraction steps were carried out on a blue-light transilluminator to minimize DNA damage, with DNA stained using GelGreen (Ozyme). Finally, DNA samples were dialyzed for 4 h on 0.025 µm nitrocellulose membranes (Milipore). 50 nM of each DNA library was then sent to MGX (CNRS Sequencing Service) for quality control and high-throughput sequencing on an Illumina Novaseq 6000 instrument using a single lane of an SP flow cell in single-read mode, producing 100 nt reads. Reads were demultiplexed by MGX using cutadapt[53]. This involved trimming the P7 adapter at the 3′ end of the reads, followed by demultiplexing based on the index located at the 3′ end and then based on the barcode at the 5′ end. The data were analyzed using the TnSeek pipeline (https://hal.science/hal-03827389), available online (https://bioinfo.cristal.univ-lille.fr/tnseek/) (supplementary Data 1; data are available at BioProject PRJNA1249410).

## Stability assay
Overnight cultures were normalized to an $OD_{600}$ of 2, then diluted 1:1000 in 5 ml of LB medium and grown at 37 °C with shaking at 160 rpm. After 8 h, the culture was diluted again 1:1000 in 5 ml of fresh LB medium and grown for an additional 16 h under the same conditions. This process was repeated for two more consecutive days. To monitor plasmid loss, samples were taken at each dilution step, and cells were plated on non-selective LB-agar plates. After overnight incubation at 37 °C, one hundred clones were streak on non-selective or Ap- supplemented LB-agar plates to determine the presence of the pOXA-48 plasmid. Plasmid stability was calculated as the percentage of colonies resistant to Ap relative to the total number of colonies tested.

## Live-cell microscopy experiments
Overnight cultures of recipient and donor cells grown separately in LB at 37 °C were diluted to $OD_{600} \sim 0.05$ in RDM and grown further at 37 °C to $OD_{600} \sim 0.5$. Then, 25 µl of donor and 75 µl recipient cultures were mixed and 50 µl dropped on a filter (MF Membrane Filters-0.45 µm HA-Ref HAWP02500 Millipore) placed on LB-agar medium. Conjugation mixes were incubated for 4 h, then resuspended in 1 ml RDM and loaded to a quasi-2D microfluidic chamber (model B04A, ONIX, CellASIC®) with pressure 5 psi for 1 min. The temperature of the microfluidic chamber was maintained at 37 °C and cells were imaged every 10 min for 3 h with the flow of fresh medium supplemented with Ap.

## Conjugation assays
**Liquid condition.** Overnight cultures grown in LB medium of recipient and donor cells were diluted to an $OD_{600}$ of 0.05 and grown further at 160 rpm to reach an $OD_{600}$ 0.7–0.9. 25 µl of donor and 75 µl of recipient cultures were mixed into an Eppendorf tube and incubated for 90 min at 37 °C. 1 ml of LB was added gently, and the tubes were incubated again for 90 min at 37 °C. Conjugation mixes were vortexed, serial diluted, and plated on LB-agar supplemented with antibiotics to select recipients and transconjugants.

**Filter condition.** Overnight cultures grown in LB medium of recipient and donor cells were normalized to an $OD_{600} = 2$. 100 µl of donor and 100 µl recipient were mixed and 100 µl dropped on a filter (MF Membrane Filters-0.45 µm HA-Ref HAWP02500 Millipore) placed on LB-agar medium. Conjugation mix were incubated for 150 min at 37 °C, then resuspended into 1 ml of LB medium, serial diluted, and plated on LB-agar supplemented with antibiotics to select recipients and transconjugants.

## *dqlB-agrB* killing and rescue assays
For killing/rescue spot assays, cells were grown at 37 °C to $OD_{600}$ ~0.4–0.5 in LB medium supplemented with 0.4% glucose, Cm and Kan.

Cultures were then serially diluted and spotted on LB-agar plates supplemented with the same antibiotics and with either 0.4% glucose or 0.2% arabinose, and incubated overnight at 30 °C. For imaging experiments, cells were grown at 37 °C in LB medium supplemented with Cm. At $OD_{600}$ ~0.4–0.5, 0.2% arabinose was added, and cells were imaged after 240 min of induction using a Ti-E microscope (Nikon) equipped with a 100x phase contrast objective (Plan apo λ 1.45, Nikon). Conjugation of TAPs was performed as previously described in liquid condition[16]. Viable counts for transconjugants were determined by plating cells on LB-agar medium supplemented with St and Kan.

## Estimation of plasmid copy number by digital PCR quantification (dPCR)
Quantifications of DNA loci: *terC* (*E. coli* chromosome) and *traN* (pOXA-48) were performed as described in[54] using multiplex dPCR (Stilla Technologies). dPCR was performed directly on cell lysate: 1 ml of overnight culture (liquid) or a scratch of colonies (solid) was washed twice with 1 ml PBS, pellets were then frozen at −20 °C and resuspended in 200 µl PBS, samples were then boiled for 10 min. Samples were centrifuged for 10 min, supernatant was recovered, and DNA content was measured using Qubit device (ThermoFisher). PCR reactions were performed with 0.1 ng of DNA using the naica® multiplex PCR MIX on a Sapphire or Ruby chip (Stilla Technologies). dPCR was conducted on a Naica Geode, and Image acquisition on the Naica Prism6 reader. Images were analyzed by Crystal Miner software (Stilla Technologies). The dPCR run was performed using the following steps: droplet partition (40 °C, atmospheric pressure AP to + 950 mbar, 12 min), initial denaturation (95 °C, +950 mbar, for 2 min), followed by 45 cycles at (95 °C for 10 s and 58 °C for 30 s), droplet release (down 25 °C, down to AP, 33 min).

## Exclusion index assays
Conjugation assays were performed between a donor strain carrying the wild-type pOXA-48 and a recipient strain without a plasmid or harboring the plasmid to be tested. The exclusion index was calculated by comparing the frequency of plasmid acquisition by a recipient strain to the frequency obtained with the plasmid-free recipient strain[33].

## Bioinformatic analysis
A dataset of 15,984 plasmid sequences with defined incompatibility groups was subsampled from the PLSDB database[26,27]. Homologs of the *dqlB* nucleotide sequences were identified using the HMMER web server[55], starting with a seed set of six known *dqlB*[22] and *dqlB* of pOXA-48. The distribution of *dqlB* homologs across plasmid incompatibility groups was determined using PLSDB metadata. Identical homologous sequences were clustered into 178 groups before alignment with the L-INS-I method of MAFFT[56]. Multiples sequence alignments were trimmed using BMGE[57], and maximum likelihood phylogenetic trees were constructed with IQ-TREE[58]. 1000 replicates ultrafast bootstrap alignments were performed. The resulting tree was visualized using TreeViewer[59], with the distribution of incompatibility group indicated for each leaf.

## Alphafold analysis
To model proteins, full-length amino acid sequences were uploaded to the Alphafold 3.0 web server[60]. Based on the published structures of DotN[30] and the T4CC[31] of *L. pneumophila*, Orf38 was modeled in the presence of a Zn ion as co-factor. The protein structures were aligned and visualized using PyMol (v1.20).

## Statistics & Reproducibility
No statistical method was used to predetermine sample size. No data were excluded from the analyses.

**Reporting summary**

Further information on research design is available in the Nature Portfolio Reporting Summary linked to this article.

## Data availability

All data to understand and assess the conclusions of this research are available in the main text, Supplementary Information and Supplementary Data 1. Source data are provided with this paper in the Source Data file. Source data are provided with this paper.

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

## Acknowledgements

This work was supported by the ANR through DeCa-P [grant number ANR-22-CE35-0017 to S.B., E.G., Y.M. and J.B.] and ETA-2 [grant number ANR-22-CE12-0032 to C.L. and N.F.], the FRM (Foundation for Medical Research) [grant number EQU202103012587 to C.L.], ANR JCJC [grant number ANR-19CE12-0001 to M.E.V.] and the University of Lyon through funding to Y.B.

## Author contributions

S.B. Conceived, designed and supervised the execution of the study; Y.B., N.F., Y.M., J.B., M.E.V., A.D.B., S.B. and E.G. performed and analyzed the data. J.D., B.I., and P.B. sequenced the pOXA-48 plasmid. S.B., C.L., E.G. and M.E.V. provided funding.

## Competing interests

The authors declare no competing interests.
