## [Transparent Peer Review file · Nature Communications]

Genetic Determinants of pOXA-48 Plasmid Maintenance and Propagation in *Escherichia coli*

Corresponding Author: Dr Sarah Bigot

Version 0:

Reviewer comments:

Reviewer #1

(Remarks to the Author)

In this study, the authors sequenced the IncL plasmid pOXA-48, which constitutes a significant health risk as a primary vehicle for carbapenem resistance in Enterobacteriaceae. Using Tn-Seq, the authors then thoroughly investigate the contributions of encoded genes to plasmid maintenance/stability, copy number, and conjugative transfer. Besides identifying genes with predicted roles in these processes, the authors identified several new determinants whose products were experimentally shown to contribute to plasmid maintenance or propagation. Using a combination of genetic, microscopy, bioinformatics, and structural modeling approaches, the authors identified functions for a novel orf (orf20) in plasmid stability, confirmed that DqIB-agrB functions as a toxin-antitoxin addiction module and determined that this module is widely distributed among many plasmids of different incompatibility groups, and presented modeling evidence that another orf (orf38) encodes a DotN structural homolog that likely interacts with the DotL and DotM homologs TrbC and TrbA, respectively. Finally, the authors explored the contributions of genes shown to be enriched in transposon insertions, which confirmed the role of excA in entry exclusion. pOXA-48 is 95.6 % identical to pCTX-M3, which so far has been more extensively characterized than pOXA-48. Additionally, a recent CRISPR interference approach was used to characterize functions of pOXA-48 genes, which resulted in identification of several previously uncharacterized genes identified in the present study, although follow-up studies in the two manuscripts did not always yield the same conclusions regarding gene functions. Overall, this study presents an extensive analysis of pOXA-48A genetic composition, highlighting previously known and new genes of importance for plasmid maintenance and propagation. By its nature, however, Tn-Seq generates only superficial information regarding underlying mechanisms of action of the few novel genes identified and their products. Accordingly, the manuscript presents as a thorough survey of pOXA-48 genetic composition, but without much new insight into how newly identified factors functionally or structurally integrate with the other well-characterized plasmid maintenance or transfer systems. Besides this general concern, the authors should address the following points, most fairly minor:

1. L. 40. Verbiage. Perhaps “kills plasmid-cured cells...”
2. L. 47. I'm not sure a plasmid feels the need to persist in diverse bacterial hosts. Reword.
3. L. 73. Define TA here.
4. L. 64. Given the close sequence relatedness of pOXA-48 with pCTX-M3, it's clear that the bulk of the annotation of pOXA-48 derives from that previously generated for pCTX-M3. This diminishes the novelty of the sequence analysis, although it's evident that the sequence had to be determined prior to the Tn-Seq.
5. L. 89. Regarding the Tn-Seq, most of the low-frequency insertions are in genes whose functions are known and have been extensively characterized, so results fit expectations and no new information is presented. Examples include the rep genes, korC, parA/B, nuc (although the role of nuc seems to differ in the two plasmids), and orf20 (recently shown by Calvo-Vilamanan).
6. L. 170. The extended studies of DqIB-agrB confirming its role as an addiction module are of interest and add significantly to our knowledge of this new and widely distributed TA system. It would be straightforward, however, to decipher the importance of the observed sequence homology between the 5' UTR of dqIB mRNA and agrB RNA to test the proposed base-pairing model.
7. L. 288. These studies are interesting, but the main information of interest pertains to ExcA, yet ExcA homologs encoded by R64 and R621a have been characterized in considerably more detail. So, fundamentally, the present work confirms that ExcA functions in entry exclusion as predicted by the previous study.
8. L. 373. This is a bit confusing, the recombination event replaced the stated genes with what? excA, traX, and traY are all present on pOXA-48, please explain.

9. L. 408. This section should be revised to better define similarities and differences in the findings reported by Calvo-Villamanan and this study; as written, it's really not clear specifically where the findings diverge, or where this study advances information beyond that reported by the Calvo-Villamanan study.

10. Table S1. There's no potential pilin gene identified, could the authors speculate on which Orf codes for the pilin subunit?

Reviewer #2

(Remarks to the Author)

The manuscript by Baffert et al. describes the use of random transposon mutagenesis and subsequent screening through high-throughput sequencing of mutant pools (Tn-seq) to identify genes that are involved in the replication, maintenance and conjugation of the plasmid pOXA-48. This plasmid is of significant public health concern as it confers resistance to carbapenems, and understanding the function of genes on the plasmid is thus of significant interest to better understand the mechanistic underpinnings of its current rapid global dissemination among Gram-negative opportunistic pathogens.

The work is carried out and presented to a high standard. Some of the functions of the identified genes are perhaps not particularly surprising (e.g. the rep genes involved in replication), but important novel insights are generated, including the identification of a novel toxin-antitoxin system and regulators of conjugation.

My comments are generally of a minor nature.

Data availability: none of the sequencing data appears to have been made available through NCBI or the European Nucleotide Archive. The authors will need share their data through these databases.

Fig. 5 and 6: it appears that these tests have not been corrected for multiple testing. I believe Dunn's post-hoc test would be appropriate here, but I would advise the authors to consult with a statistician.

L. 233 (or elsewhere): it would aid in the interpretation of the data if conjugation efficiencies of pOXA-48 under the conditions used here would be added to the manuscript. Is the efficiency sufficiently high to allow representation of all transposon mutants in the dataset?

I. 243: is orf36.2 a typo here? Elsewhere only orf36.1 appears to be discussed.

I. 247: it would have been of interest to measure expression levels (through qPCR) of the conjugation module to provide further evidence on the function of orf36.1 and orf38, particularly if they act as transcriptional regulators.

I. 321: add reference after statement 'harbor the pOXA-48 plasmid'

I. 518: provide more information (specifically read-length) on Illumina sequencing methodologies.

Suggested changes in the manuscript text

L. 38: write 'poorly understood'.

L. 40: write 'kill cells that have lost the plasmid'

L. 96: write 'translationally coupled'

I. 112: write 'uniquely'

L. 221: write 'Each leave was'

I. 230: write 'is currently likely overlooked'

Reviewer #3

(Remarks to the Author)

The manuscript by Baffert and colleagues presents a body of work identifying the genetic determinants of plasmid maintenance and propagation for the IncL plasmid pOXA-48. This body of work is very comprehensive and well written. The authors have used a combination of TraDIS and experimental validation (stability assays, conjugations assays, deletions and complementation) to demonstrate roles for over thirty genes in functions including stability conjugative transfer and entry-exclusion. The authors also accounted for both liquid and filter conjugation.

It is noted that a contemporaneous study recently published by Calvo-Villamanan and colleagues has significant overlap with this body of work. However, I do not see this as a major issue. The studies are complementary to one another, and some of the differences arising from the different approaches (CRISPR silencing vs TraDIS) provide important insights into the roles of the genes identified.

I only have minor comments that I believe will improve the manuscript.

- Line 73. Should be "aligned to **the** pOXA-48 plasmid".

- In lines 240-247 there is some inconsistency in the gene/protein being referred to. I think the issue is line 243, where it refers to "orf36.2". From what I can tell, it should be "orf36.1".

Version 1:

Reviewer comments:

Reviewer #1

(Remarks to the Author)

The authors have addressed my prior concerns, in part by highlighting novel findings in relation to previously published work and the recent preprint by Calvo-Vilamanan. Additionally, they performed experiments to validate the proposed RNA base-pairing model for *dqIB-agrB*; these data are now included. This study comprehensively identifies important regulatory features contributing to *pOXA-48* spread and maintenance, the medical significance of which is underscored by the role of these plasmids in dissemination of carbapenem resistance. This reviewer has no further concerns about this nice contribution to the field of plasmid biology.

Reviewer #2

(Remarks to the Author)

The authors have convincingly addressed the comments of the reviewers.

Reviewer's Comments:

Reviewer #1 (Remarks to the Author)

In this study, the authors sequenced the IncL plasmid pOXA-48, which constitutes a significant health risk as a primary vehicle for carbapenem resistance in Enterobacteriaceae. Using Tn-Seq, the authors then thoroughly investigate the contributions of encoded genes to plasmid maintenance/stability, copy number, and conjugative transfer. Besides identifying genes with predicted roles in these processes, the authors identified several new determinants whose products were experimentally shown to contribute to plasmid maintenance or propagation. Using a combination of genetic, microscopy, bioinformatics, and structural modeling approaches, the authors identified functions for a novel orf (orf20) in plasmid stability, confirmed that DqIB-agrB functions as a toxin-antitoxin addiction module and determined that this module is widely distributed among many plasmids of different incompatibility groups, and presented modeling evidence that another orf (orf38) encodes a DotN structural homolog that likely interacts with the DotL and DotM homologs TrbC and TrbA, respectively. Finally, the authors explored the contributions of genes shown to be enriched in transposon insertions, which confirmed the role of *excA* in entry exclusion. pOXA-48 is 95.6 % identical to pCTX-M3, which so far has been more extensively characterized than pOXA-48. Additionally, a recent CRISPR interference approach was used to characterize functions of pOXA-48 genes, which resulted in identification of several of several previously uncharacterized genes identified in the present study, although follow-up studies in the two manuscripts did not always yield the same conclusions regarding gene functions. Overall, this study presents an extensive analysis of pOXA-48A genetic composition, highlighting previously known and new genes of importance for plasmid maintenance and propagation. By its nature, however, Tn-Seq generates only superficial information regarding underlying mechanisms of action of the few novel genes identified and their products. Accordingly, the manuscript presents as a thorough survey of pOXA-48 genetic composition, but without much new insight into how newly identified factors functionally or structurally integrate with the other well-characterized plasmid maintenance or transfer systems.

> We thank the reviewer for this comprehensive summary and for recognizing the thorough survey of pOXA-48 genetic composition.

Below, we provide our detailed point-by-point responses to the reviewer's comments.

Besides this general concern, the authors should address the following points, most fairly minor:

1. L. 40. Verbiage. Perhaps "kills plasmid-cured cells..."

> This has been corrected by 'kill cells that have lost the plasmid' as suggested by the reviewer 2.

2. L. 47. I'm not sure a plasmid feels the need to persist in diverse bacterial hosts. Rework.

> The sentence has been reworded to avoid anthropomorphic language and better reflect the biological context. We state now line 46–47: "Understanding these mechanisms is crucial to determining how pOXA-48 balances the cost of transfer while ensuring its persistence across diverse bacterial hosts."

3. L. 73. Define TA here.

> We thank the reviewer for this comment and understand that the abbreviation "TA" may have caused some confusion with a toxin-antitoxin (TA) system. In the sentence at line 73, "TA" refers to the TA dinucleotide recognized by the Himar-1 transposon. To clarify this point and avoid ambiguity, we have modified the text line 77 to explicitly read "TA dinucleotide."

4. L. 64. Given the close sequence relatedness of pOXA-48 with pCTX-M3, it's clear that the bulk of the annotation of pOXA-48 derives from that previously generated for pCTX-M3. This diminishes the novelty of the sequence analysis, although it's evident that the sequence had to be determined prior to the Tn-Seq.

> The reviewer is correct in noting that pOXA-48 shares high nucleotide identity with pCTX-M3. This sequence similarity has indeed been used to predict the genetic organization of pOXA-48 in previous studies. However, no comprehensive experimental work to validate these predictions has been carried out to date. Our study goes beyond annotation by functionally characterizing several key genetic elements, including novel factors not previously described in pCTX-M3. Moreover, our Tn-Seq-based approach allowed us to identify new determinants of plasmid maintenance and transfer, and to uncover functional differences between homologous genes in pOXA-48 and pCTX-M3, such as the distinct phenotypes observed for the *excA* or *nuc* deletion. Thus, while the sequence relationship provided a framework, our findings represent a significant advance in understanding the molecular biology of pOXA-48.

5. L. 89. Regarding the Tn-Seq, most of the low-frequency insertions are in genes whose functions are known and have been extensively characterized, so results fit expectations and no new information is presented. Examples include the *rep* genes, *korC*, *parA/B*, *nuc* (although the role of *nuc* seems to differ in the two plasmids), and *orf20* (recently shown by Calvo-Vilamanan).

> We agree that our study confirms previously predicted roles for the *rep* and *parAB* genes, and we acknowledge that these results support prior expectations. However, as the reviewer also notes, we found that the function of *nuc* differs between the pOXA-48 and pCTX-M3 plasmids. This underlines the importance of experimentally re-evaluating genes with predicted functions, as they may carry divergent roles in different plasmid contexts.

We would like to highlight that only a few studies have investigated *korC* gene in detail, specifically in the RP4 plasmid (Kornacki *et al.* 1990, PMID: 2160936; Kornacki *et al.* 1993, PMID: 8349548; Goncharoff *et al.*, PMID: 2045366) and IncU plasmids (Ludwiczak *et al.*, PMID: 23583562). These studies describe KorC as a putative transcriptional regulator that works in coordination with KorA to regulate particularly the expression of *klcA* gene also present on the pOXA-48 plasmid. In our study, we show that KorC in pOXA-48 exhibits distinct features: (1) only a single KorC binding site is predicted, in contrast to the multiple sites found in other plasmids; (2) this putative binding site is located upstream *orf24* and not upstream of *klcA*; and (3) pOXA-48 does not encode a KorA homolog, suggesting a regulatory mechanism independent of KorA. These findings represent new insights into KorC function in the context of IncL plasmids, which likely differs from other plasmids.

Regarding *orf20*, we respectfully disagree with the statement that this gene has been extensively characterized. This gene has only recently been identified in a single study by Calvo-Villamañán *et al.*, which was submitted in parallel to ours. We collaborated with Dr. Alvaro San Millán, corresponding author of that study, and jointly decided to deposit our manuscripts on bioRxiv and submit them back-to-back to Nature Communications, as our results were complementary. Orf20 has no identifiable domains or conserved motifs, and until now, its essential role in plasmid stability had not been documented in any plasmid system. We believe that the discovery and functional validation of Orf20 as a stability determinant constitutes a key original contribution of our work.

6. L. 170. The extended studies of DqIB-*agrB* confirming its role as an addiction module are of interest and add significantly to our knowledge of this new and widely distributed TA system. It would be straightforward, however, to decipher the importance of the observed sequence homology between the 5' UTR of *dqIB* mRNA and *agrB* RNA to test the proposed base-pairing model.

> As suggested by the reviewer, we have experimentally performed additional experiments, which demonstrate the importance of base-pairing interaction between *agrB* and the 5' UTR of *dqIB*. To do this, we generated an *agrB* mutant carrying a 4-nt mismatch that disrupted the predicted pairing, as well a compensatory *dqIB* mutant in which base-pairing was restored. The results are now presented in a new Supplementary Figure S2 and are described in the revised manuscript line 195-203: "To determine whether repression of *dqIB* by *agrB* relies on a complementary base-pairing mechanism between the *agrB* RNA and the 5' UTR of *dqIB*, we constructed an *agrB* mutant with a 4-nt mismatch in the predicted pairing region (Figure S2a). This mutant failed to repress the toxicity induced by wild-type DqIB (Figure S2c). We then generated a compensatory mutation into the 5' UTR of *dqIB*, restoring complementarity to the mutated *agrB*. Reestablishing base pairing fully restored the ability of the *agrB* mutant to neutralize DqIB toxicity (Figure S2c). These results demonstrate that specific base-pairing interactions are critical for *agrB* to function as an effective antitoxin."

7. L. 288. These studies are interesting, but the main information of interest pertains to ExcA, yet ExcA homologs encoded by R64 and R621a have been characterized in considerably more detail. So, fundamentally, the present work confirms that ExcA functions in entry exclusion as predicted by the previous study.

> We fully agree that *excA* homologs, such as those from the R64 and R621a plasmids, have been characterized in detail and that previous studies predicted a role for *excA* in entry exclusion. In the present work, we experimentally confirm that the *excA* gene annotated in the pOXA-48 plasmid indeed functions as an entry exclusion factor, explaining the strong enrichment of transposon insertions observed in *excA* during our Tn-Seq screen.

However, our data also go beyond this validation by providing additional insights into ExcA biology that, to our knowledge, have not been reported for its homologs. Notably, we found that *excA* deletion leads to a significant increase in plasmid copy number (PCN). To investigate the biological consequences of this PCN elevation, we performed a new experiment (now presented in Figure 6f and Line 366-377) in which we assessed conjugation frequency using donor cells collected directly from colonies grown on solid medium. In this condition, which correlates with high PCN, *excA* mutants displayed a 12-fold increase in conjugation frequency compared to the wild type. This phenotype was reversed by ectopic *excA* expression, restoring conjugation to wild-type levels. These findings establish a direct link between *excA*, PCN, and conjugation efficiency, a phenotype not previously described for other *excA* homologs. This again explains the strong enrichment of *excA* mutants in our Tn-Seq output library.

In addition, our work highlights the value of re-examining genes with predicted functions, as experimental validation can reveal distinct roles. This is illustrated by the contrasting transfer phenotypes observed for *excA* mutants of pOXA-48 and pCTX-M3 plasmids. While deletion of *excA* in pCTX-M3 was shown to reduce plasmid transfer, our results reveal that *excA* deletion in pOXA-48 unexpectedly leads to a substantial increase in conjugation frequency under specific conditions. Together, these findings support the relevance and novelty of our results, extending beyond previously predicted functions.

8. L. 373. This is a bit confusing, the recombination event replaced the stated genes with what? *excA*, *traX*, and *traY* are all present on pOXA-48, please explain.

> We thank the reviewer for pointing this out and agree that the sentence may have been unclear in its current form. We have now added a paragraph (line 404-413) that states: "The *traX*, *traY*, and *excA* genes of the pOXA-48 and pCTX-M3 plasmids are more divergent from one another than other genes, encoding proteins with significantly lower amino acid identity (Table S1). While the exact origin of this divergence remains unknown, one hypothesis is

that a recombination event occurred during the evolutionary history of these plasmids. It has been proposed that both pOXA-48 and pCTX-M3 derived from the ancestral pEL60 plasmid. Interestingly, pCTX-M3 harbors *traX*, *traY*, and *excA* genes that are nearly identical to those of pEL60, whereas the corresponding genes in pOXA-48 have substantially diverged. This suggests that a gene exchange event may have introduced a variant region in the pOXA-48 lineage, replacing the original gene cluster found in the ancestral plasmid.”

9. L. 408. This section should be revised to better define similarities and differences in the findings reported by Calvo-Villamanan and this study; as written, it's really not clear specifically where the findings diverge, or where this study advances information beyond that reported by the Calvo-Villamanan study.

> We agree with the reviewer that the original paragraph lacked clarity in defining the similarities and differences between our study and that of Calvo-Villamañán *et al.* We have now rewritten this section. The revised paragraph states now line 451-460 : “The essentiality of *orf20* and *korC* was also reported by Calvo-Villamañán *et al.*, who used a plasmid-wide CRISPRi screen to investigate pOXA-48 fitness effects in various clinical strains of enterobacteria. While our experiments were conducted in the laboratory strain *E. coli* K12 MG1655, their study included both clinical isolates and the laboratory strain *E. coli* J53. They showed that silencing *pemI*, *pri* and H-NS (*orf36.2*) in J53 had minimal effects, consistent with our findings in MG1655, where no significant decrease in transposon insertions was observed for these genes. In contrast, expression of these genes is essential in clinical isolates, highlighting the importance of plasmid-host interactions and supporting the idea that gene essentiality is context-dependent, influenced by both genetic and physiological characteristics of the host strain.”

10. Table S1. There's no potential pilin gene identified, could the authors speculate on which Orf codes for the pilin subunit?

> We thank the reviewer for this interesting question. The pOXA-48 plasmid encodes a mating pair formation system of the MPF_I type, for which IncI plasmids serve as prototypes. To date, the specific gene encoding the conjugative pilin subunit in this system has not been clearly identified or experimentally validated. However, the study by Guglielmini *et al.* (PMID: 24623814) showed that TraR and TraQ from MPF_I systems, which are homologous, share an indirect evolutionary relationship with the pilin protein VirB2. Given that this prediction has not, to our knowledge, been experimentally confirmed, we chose not to annotate these genes as pilin candidates in Table S1.

Reviewer #2 (Remarks to the Author)

The manuscript by Baffert *et al.* describes the use of random transposon mutagenesis and subsequent screening through high-throughput sequencing of mutant pools (Tn-seq) to identify genes that are involved in the replication, maintenance and conjugation of the plasmid pOXA-48. This plasmid is of significant public health concern as it confers resistance to carbapenems, and understanding the function of genes on the plasmid is thus of significant interest to better understand the mechanistic underpinnings of its current rapid global dissemination among Gram-negative opportunistic pathogens.

The work is carried out and presented to a high standard. Some of the functions of the identified genes are perhaps not particularly surprising (e.g. the rep genes involved in replication), but important novel insights are generated, including the identification of a novel toxin-antitoxin system and regulators of conjugation.

> We thank the reviewer for the positive feedback and for recognizing the relevance of our work.

My comments are generally of a minor nature.

Data availability: none of the sequencing data appears to have been made available through NCBI or the European Nucleotide Archive. The authors will need share their data through these databases.

> The pOXA-48 sequencing and Tn-seq sequencing data are now available through the BioProject PRJEB88520 and PRJNA1249410 respectively. These informations have been added to the material and methods section.

Fig. 5 and 6: it appears that these tests have not been corrected for multiple testing. I believe Dunn's post-hoc test would be appropriate here, but I would advise the authors to consult with a statistician.

> We thank the reviewer for this valuable comment regarding the statistical analysis. We agree that multiple comparisons require appropriate statistical correction to avoid overestimating significance. Performing multiple pairwise comparisons using uncorrected tests like the t-test increases the risk of false positives, simply due to the number of tests performed. To address this, we repeated and expanded the number of replicates for some conditions and reanalyzed the data using a Kruskal-Wallis test followed by Dunn's post-hoc test, which is well suited for non-parametric data and allows for multiple comparisons. In our analysis, each condition was specifically compared to the wild-type, and the updated statistical results are now reflected in Figures 5 and 6 and their corresponding legends.

Using this updated statistical analysis, one condition is no longer significantly different. Specifically, the conjugation frequency measured for the *orf38* mutant complemented with Orf38 remains significantly lower than that of the wild type. However, complementation still resulted in a 10⁹-fold increase in conjugation frequency compared to the mutant, validating its functional effect, although partial. We now explain this in the revised version, line 262-266

« The ectopic production of Orf36.1 from a complementation plasmid fully restored conjugation frequency to the wild-type levels and production of Orf38 in the *orf38* mutant led to a 10⁵-fold increase in conjugation frequency. However, this complementation remained partial, as it did not fully restore the transfer frequency to that of the wild-type.»

L. 233 (or elsewhere): it would aid in the interpretation of the data if conjugation efficiencies of pOXA-48 under the conditions used here would be added to the manuscript. Is the efficiency sufficiently high to allow representation of all transposon mutants in the dataset?

> We thank the reviewer for raising this important point. The conjugation frequency of the pOXA-48 mutant library was approximately 0.3%, as determined based on the ratio of transconjugants to total recipient cells prior to library output preparation. Although this value reflects a relatively low frequency of plasmid transfer, we collected approximately 2 million transconjugant colonies. Given the 2,788 TA dinucleotide sites present on the pOXA-48 plasmid, this represents an average coverage of ~715 transconjugants per unique insertion site, assuming uniform transfer. While not all mutants are expected to transfer equally, this depth of coverage ensures that the vast majority of transferable insertion mutants were robustly represented in the dataset, supporting the robustness of our Tn-Seq analysis despite the low conjugation rate.

We have added a sentence in the Materials and Methods section to clarify this point, line 581-586: “After overnight incubation at 37°C, approximately 2,000,000 transconjugant colonies were collected, providing an estimated ~715-fold theoretical coverage of the 2,788 potential TA insertion sites in the pOXA-48 plasmid. This ensured robust representation of the mutant library in the output pool, despite the relatively low conjugation frequency (~0.3%). The resulting output library was resuspended in LB medium and frozen in 40% glycerol at -80°C.”

I. 243: is *orf36.2* a typo here? Elsewhere only *orf36.1* appears to be discussed.

> We thank the reviewer for pointing this out, as also noted by Reviewer 3. This was indeed a typographical error, which has now been corrected to *orf36.1* in the revised manuscript line 258.

I. 247: it would have been of interest to measure expression levels (through qPCR) of the conjugation module to provide further evidence on the function of *orf36.1* and *orf38*, particularly if they act as transcriptional regulators.

> We thank the reviewer for suggesting that investigation of the regulatory mechanisms of Orf36.1 would provide further insight into its role. While our study identified Orf36.1 as a candidate transcriptional regulator based on domain predictions, we chose not to include additional functional characterization in the current manuscript, as this work is ongoing in our lab. Specifically, we are currently investigating the regulon of Orf36.1 using both global (e.g., RNA-seq) and targeted approaches, including qPCR, as suggested by the reviewer. These analyses will be part of another separate manuscript.

Regarding Orf38, structural predictions and functional modeling do not suggest a role in transcriptional regulation. Instead, our data support its involvement as a structural component of the type IV coupling complex (T4CC), where it likely functions alongside TrbC (a DotL/VirD4 homolog) and TrbA (DotM homolog), potentially participating in the DNA transfer process itself.

I. 321: add reference after statement 'harbor the pOXA-48 plasmid'

> This statement is based on the predicted homology of *excA* with known exclusion genes. To our knowledge, no prior study has experimentally confirmed this function for the *excA* gene specifically in the pOXA-48 plasmid. Our study provides direct evidence supporting and demonstrating this predicted role in the context of pOXA-48, which is why no reference is cited for this statement in the manuscript.

I. 518: provide more information (specifically read-length) on Illumina sequencing methodologies.

> We have added a paragraph line 609-616 stating: “50nM of each DNA library was then sent to MGX (CNRS Sequencing Service) for quality control and high-throughput sequencing on Illumina Novaseq 6000 instrument using a single lane of an SP flow cell in single-read mode, producing 100 nt reads. Reads were demultiplexed by MGX using cutadapt. This involved trimming the P7 adapter at the 3' end of the reads, followed by demultiplexing based on the index located at the 3' end and then based on the barcode at the 5' end. The data were analyzed using the TnSeek pipeline (<https://hal.science/hal-03827389>), available online (<https://bioinfo.cristal.univ-lille.fr/talseek/>).

Suggested changes in the manuscript text

L. 38: write 'poorly understood'. Corrected

L. 40: write 'kill cells that have lost the plasmid'. Corrected

L. 96: write 'translationally coupled'. Corrected

I. 112: write 'uniquely'. Corrected

L. 221: write 'Each leave was'. Corrected

I. 230: write 'is currently likely overlooked'. Corrected

Reviewer #3 (Remarks to the Author):

The manuscript by Baffert and colleagues presents a body of work identifying the genetic determinants of plasmid maintenance and propagation for the IncL plasmid pOXA-48. This body of work is very comprehensive and well

written. The authors have used a combination of TraDIS and experimental validation (stability assays, conjugations assays, deletions and complementation) to demonstrate roles for over thirty genes in functions including stability conjugative transfer and entry-exclusion. The authors also accounted for both liquid and filter conjugation.

It is noted that a contemporaneous study recently published by Calvo-Villamanan and colleagues has significant overlap with this body of work. However, I do not see this as a major issue. The studies are complementary to one another, and some of the differences arising from the different approaches (CRISPR silencing vs TraDIS) provide important insights into the roles of the genes identified.

> We sincerely thank the reviewer for the positive comments on our work. We also appreciate his/her recognition that the recently published study by Calvo-Villamañán and colleagues is not a concern regarding overlap. As the two studies provide complementary data, we had agreed in coordination with Alvaro San Millán (corresponding author of that study) to coordinate the release of both manuscripts by posting them simultaneously on bioRxiv and to submitting them back-to-back to Nature Communications.

I only have minor comments that I believe will improve the manuscript.

- Line 73. Should be "aligned to **the** pOXA-48 plasmid". Corrected

- In lines 240-247 there is some inconsistency in the gene/protein being referred to. I think the issue is line 243, where it refers to "orf36.2". From what I can tell, it should be "orf36.1".

>We thank the reviewer for pointing this out, as also noted by Reviewer 2. This was indeed a typographical error, which has now been corrected to *orf36.1* in the revised manuscript line 258.